# A KWS System for Edge-Computing Applications with Analog-Based Feature Extraction and Learned Step Size Quantized Classifier

**DOI:** 10.3390/s25082550

**Published:** 2025-04-17

**Authors:** Yukai Shen, Binyi Wu, Dietmar Straeussnigg, Eric Gutierrez

**Affiliations:** 1Electronics Technology Department, University of Madrid Carlos III, 28911 Leganes, Spain; eric.gutierrez@uc3m.es; 2Independent Researcher, 01187 Dresden, Germany; wbybinyiwu@gmail.com; 3Power and Sensor Systems, Infineon Technologies, 9500 Villach, Austria; dietmar.straeussnigg@infineon.com

**Keywords:** keyword spotting (KWS), quantization-aware training (QAT), analog feature extraction, recurrent neural network (RNN), edge computing

## Abstract

Edge-computing applications demand ultra-low-power architectures for both feature extraction and classification tasks. In this manuscript, a Keyword Spotting (KWS) system tailored for energy-constrained portable environments is proposed. A 16-channel analog filter bank is employed for audio feature extraction, followed by a digital Gated Recurrent Unit (GRU) classifier. The filter bank is behaviorally modeled, making use of second-order band-pass transfer functions, simulating the analog front-end (AFE) processing. To enable efficient deployment, the GRU classifier is trained using a Learned Step Size (LSQ) and Look-Up Table (LUT)-aware quantization method. The resulting quantized model, with 4-bit weights and 8-bit activation functions (W4A8), achieves 91.35% accuracy across 12 classes, including 10 keywords from the Google Speech Command Dataset v2 (GSCDv2), with less than 1% degradation compared to its full-precision counterpart. The model is estimated to require only 34.8 kB of memory and 62,400 multiply–accumulate (MAC) operations per inference in real-time settings. Furthermore, the robustness of the AFE against noise and analog impairments is evaluated by injecting Gaussian noise and perturbing the filter parameters (center frequency and quality factor) in the test data, respectively. The obtained results confirm a strong classification performance even under degraded circuit-level conditions, supporting the suitability of the proposed system for ultra-low-power, noise-resilient edge applications.

## 1. Introduction

With the advancement of portable devices and artificial intelligence (AI) techniques, the demand for energy-efficient processing of sensor data on the edge has increased rapidly. Edge computing offers significant benefits over remote cloud-based processing, including reduced data transmission bandwidth, lower latency, and enhanced sensitive data privacy at the expense of extremely restricted power requirements [1]. Energy efficiency is critical for edge computing due to the constraints of limited battery capacity or reliance on energy harvesting circuits. This usually leads to simpler classification and regression circuit architectures with weaker pattern recognition capabilities than complex algorithms executed in CPU/GPUs. Ultra-low power structures are then required to enlarge the number of processing elements and achieve sufficiently high accuracy and resolution performance [2].

A foundational edge-AI application is Keyword Spotting (KWS), widely utilized in human–machine interaction scenarios such as virtual reality or automatic speech recognition (ASR). Acting as a power-gating mechanism, KWS triggers downstream function blocks only when a command word is detected, enabling voice-activated devices. In the same context, Voice Activity Detection (VAD) systems are simplified versions of KWS architectures designed only to distinguish between whether a sound corresponds to human voice or not [3,4]. The always-on and smart nature of either VAD or KWS systems requires a trade-off between classification simplicity and performance, and power efficiency.

Figure 1 depicts the conventional architecture of a KWS system. It is composed of an analog front-end (AFE) that performs filtering, amplification and digitization operations, and a digital back-end that performs intensive machine learning computation to extract data features and classify them [5,6]. While this approach often leads to excellent classification performance, it may not be suitable for the requirements of edge computing due to the power-hungry nature of the solution. In an attempt to reduce such power consumption, many works have focused on carrying typically digital-based operations to the analog domain, where they might occupy more silicon area, but show much higher energy efficiency. Early efforts in mixed-signal pattern recognition systems, especially in focal-plane vision chips, have demonstrated the potential of analog pre-processing for low-power computation. A comprehensive overview of such approaches is summarized in [7], which systematically examines the benefits and inherent limitations of AFE-based processing in intelligent sensing systems. These pioneering works highlight that while analog processing can offer substantial energy efficiency, it often comes at the cost of increased silicon area, reduced flexibility, and sensitivity to fabrication mismatches and drift. These insights have inspired modern edge-AI architectures, particularly in audio sensing, to explore hybrid systems that leverage the energy efficiency of analog computation and the flexibility of digital processing [8,9]. Splitting up the smart computing steps into the feature extraction and the classification steps is found some works where analog-based feature extraction stages are proposed [10,11], and also those where both stages work in the analog domain [3,12]. The analog nature of the solutions enables architectures with a power consumption of less than 100 μW. Finally, a recent trend is towards neuromorphic-based systems for ultra-low power devices, both in the analog [13] and in the digital domain [14].

The current manuscript is framed within the context described above, aiming to develop energy-efficient solutions for edge-deployed KWS systems. To address the aforementioned challenges, a highly efficient KWS architecture is proposed, which incorporates a behavior-level analog filter bank for feature extraction, followed by a digital gated recurrent unit (GRU)-based classifier. In this work, the AFE is modeled using second-order band-pass filters to simulate the spectral decomposition performed, focusing on no specific circuit implementation. Excellent classification performance is achieved with reduced model complexity and hardware optimizations. In addition, to assess the system’s feasibility under real-world operating conditions, the AFE noise and impairments are evaluated by injecting Gaussian noise at various signal-to-noise ratios (SNRs) and by perturbing the analog filter parameters (center frequency $w_{0}$ and quality factor *Q*) for mismatch and process-voltage-temperature (PVT) variations. The results show that the accuracy in both the full-precision and quantized models remains around 91% even with low SNRs and moderate parameter deviations.

This work introduces several key innovations that distinguish it from prior art. The main contributions are summarized as follows:Outstanding trade-off between classification accuracy and complexity. The proposed KWS system employs the same analog feature extraction configuration as in [10], but achieves higher accuracy, 92.33% with the full-precision model and 91.35% with the quantized model, compared to 91.00% in [10]. The GRU-based classifier uses only 80 neurons per layer, versus 128 in the reference design.Learned Step Size (LSQ) and Look-Up Table (LUT)-aware quantization for efficient deployment. Unlike previous works that use post-training or fixed-point quantization, the model leverages LSQ and LUT-aware quantization during training. This enables adaptive optimization of quantization parameters, reducing memory and power consumption while maintaining high recognition accuracy. The resulting model operates with 4-bit weights and 8-bit activations (W4A8), significantly lowering computational complexity compared to models using 8-bit weights and 12-bit or higher activations in [12,15].Noise and impairments robustness analyses of the AFE. The quantized model maintains 91% accuracy at 40 dB SNR and tolerates up to a 5% relative standard deviation from the nominal parameters in filtering operations (due to mismatch and PVT), demonstrating suitable robustness for practical edge-AI systems.

The document has the structure outlined below. Section 2 shows the foundations of the KWS system, focusing on the analog-based feature extraction stage and the digital GRU-based classifier. Section 3 describes the LSQ methodology used to train the classifier stage. Section 4 presents the dataset for training and inference, and discusses the obtained results. Finally, Section 5 concludes the manuscript.

## 2. Description of the Proposed KWS Architecture

A schematic diagram of the KWS is shown in Figure 2. The system consists of a microphone as the sensing stage, a 16-channel log-spaced analog filter bank, an analog frame and energy extractor, and an ADC block with a sampling clock frequency $f_{clk}$, followed by a GRU-based digital classifier. The classifier comprises two GRU-based layers, each with 80 units, concluding with a fully connected (FC) classification layer with 12 output neurons, corresponding to the 12 classes.

### 2.1. Behavioral Modeling of the Analog Filter Bank Feature Extractor

A conventional approach to extract the audio features in the analog domain is illustrated in Figure 3. The process begins with a set of analog band-pass filters, each tuned to a different center frequency, which are connected to the raw audio input signal. Next, wave rectifiers convert the filtered signals into their envelope representations. Finally, the energy within each frame is estimated by integrating the rectified signal over the frame’s time step. In this work, the analog filter bank is behaviorally modeled with a second-order band-pass filter transfer function, simulating the energy-efficient AFE processing. The model mimics the spectral decomposition without specifying a physical circuit, focusing on evaluating downstream quantized classification performance.

The transfer function of the band-pass filters can be expressed as follows:(1)$$H\left(s\right)=\frac{\frac{w_{0}}{Q}\cdot s}{s^{2}+\frac{w_{0}}{Q}\cdot s+w_{0}^{2}},$$
where $w_{0}$ represents the center frequency of the filter, and *Q* is the quality factor. The same filter bank configuration as in [10] is used, implementing 16 log-spaced band-pass filters with the center frequencies $w_{0}$ ranging from 125 Hz to 5 kHz and a quality factor of 4.5. Figure 4 shows the frequency response of the 16 selected band-pass filters. To ensure consistent amplitude scaling, each filter output is normalized to have unitary gain at its center frequency $w_{0}$.

To mimic the analog-domain rectification and integration over a frame duration in the feature extraction process applied to the filtered signal, the energy of each frame is computed using the following equation:(2)$$E_{n}=\frac{1}{N}\sum_{i=0}^{N-1}\left|x\left[n-i\right]\right|,$$
where $E_{n}$ represents the energy of the *n*-th frame, which consists of *N* sampled points withing this frame.

Figure 5a shows an example of the extracted features from the keyword “yes”, where a 16-channel analog filter bank is used to decompose the signal into log-spaced frequency bands. The heatmap illustrates the temporal-spectral structure across the frames. As expected, low-frequency filters capture voiced phonemes, while high-frequency filters emphasize fricative components. Figure 5b provides a visualization of the band-pass filter outputs for the selected channels, demonstrating that the analog filter bank effectively decomposes speech into distinct frequency components suitable for downstream classification.

### 2.2. GRU-Based Classifier

In [16], various neural network architectures were trained and compared for KWS targeted on hardware resource-constrained microcontrollers. It was demonstrated that Recurrent Neural Network (RNN) models, including the Long-Short Term Memory (LSTM) or GRU models, outperformed their counterparts like Convolutional Neural Networks (CNNs) or standard feed-forward models in classification accuracy, memory footprint, and number of operations per inference. Although the convolutional RNN (CRNN) or depth-wise separable CNN (DS-CNN) showed even better performance than RNNs, they require access to the full receptive field of the input such as a large amount of the input audio frames, which will increase the latency due to the input window buffering. In contrast, RNN models can process one input frame at a time, maintaining an internal state across frames. This enables RNNs to process audio data in a streaming manner without requiring a large input window buffer, making them inherently suited for real-time applications where audio frames arrive sequentially [17].

The two most popular RNN variants are LSTM and GRU. They capture the temporal dependencies of the speech signals and model the time-varying nature of them with their internal gate mechanism, and mitigate the vanishing gradient problem that emerges in the vanilla RNN model when the sequence becomes longer. GRU models are simpler than LSTM ones because they have one fewer gate reducing the computing complexity, and thus are preferred in the integrated circuit community [10,12]. Additionally, there are other edge device optimized RNN variants applied in the KWS application, such as skip RNN [15,18] and Δ-GRU [14,19]. These variants are more power efficient combined with specific AFEs and digital feature extraction circuits. Given the focus of this work on evaluating the impact of LSQ-based quantization, a standard GRU architecture is adopted to provide a clean and consistent baseline. This choice allows us to isolate the effect of the quantization method with respect to the influence of dealing with simplified or modified GRU variants. More advanced or energy-efficient architectures, such as Fast-RNN [20] or Mamba [21], remain promising candidates for future exploration.

Increasing the number of units per layer and the number of layers in a GRU generally enhances the capability to capture complex patterns from input data and process them within multiple levels of abstraction. This improvement is particularly beneficial for KWS tasks involving classifying and detecting a large amount of words, phonetically similar keywords, or dealing with noisy conditions. Nevertheless, it comes at the cost of increased complexity and computational demand, posing significant challenges for their deployment on edge devices.

Figure 6 illustrates how the accuracy performance of GRU-based classifiers depends on the number of units per layer (Figure 6a and on the number of hidden layers (Figure 6b, demonstrating that, while not incurring overfitting, the performance improves with deeper GRU models. For this particular case, overfitting is observed with too complex models, as seen in Figure 6a for a number of units per layer higher than 110. Note that all other hyperparameters remain constant during training and may not be fully optimized, as will be discussed in Section 4.2. As training a model with an overall accuracy above 90% is aimed, while keeping the size as small as possible, 80 units are selected per each hidden layer and a depth of two hidden layers are selected in this work. A fully-connected layer is finally added to perform the classification task.

## 3. Learned Step Size Quantized GRU Classifier

### 3.1. Background

In [22] a comprehensive survey on the efficient compression and execution of deep neural networks from both algorithms and hardware perspectives is provided. According to the authors, quantizing RNNs is more challenging than quantizing CNNs due to the recurrent nature and internal state dependence of RNNs. Quantization errors can propagate and accumulate across multiple time steps and significantly impact performance. Complex RNN structures, such as GRU and LSTM units, rely on gating mechanisms and sophisticated non-linear operations like *sigmoid* and *tanh* functions, which are particularly sensitive to limited precision.

An integer-only quantization method was proposed for CNNs in [23]. This method was later adapted to RNNs, as shown in [17,24,25], but was limited to the post-training quantization stage (PTQ). In their approach, a real number *r* is quantized using the following equation: (3)$$r_{q}=\mathrm{clip}(\mathrm{round}\left(\frac{r-Z}{S}\right),Q_{min},Q_{max}),$$
where $r_{q}$ is the quantized value of the real number *r*, $Q_{min}$ and $Q_{max}$ denote the quantization range, respectively, and *S* and *Z* are the scale and the zero points, defined by the following: (4)$$\begin{array}{cc} S & =\frac{\left|\max(M)|+|\min(M)\right|}{2^{bw}-1}, \end{array}$$(5)$$\begin{array}{cc} Z & =\mathrm{round}\left(\frac{-\min(M)}{S}\right), \end{array}$$
where M is weight or activation tensor with any shape and bw is the total bit-width.

Another quantization method applied to RNNs is fixed-point quantization [12,14], represented in the ***Qi.f*** format, where ***i*** and ***f*** denote the number of bits before and after the decimal point, respectively. A real number *r* can be then quantized using the following equations: (6)$$\begin{array}{cc} r_{q} & =\mathrm{clip}(\mathrm{round}\left(r\ll f\right),Q_{min},Q_{max}), \end{array}$$(7)$$\begin{array}{cc} r_{dq} & =r_{q}\gg f, \end{array}$$
where ≪ and ≫ represent bit shifting operations, and $r_{q}$ and $r_{dq}$ are the quantized and de-quantized values of a real number *r*, respectively. The challenge of fixed-point quantization lies in defining the word and the integer lengths, i.e., selecting the position of the binary point for a given bit-width. This requires an in-depth analysis of the dynamic range of the weights and activation functions in an already trained full-precision model. Based on this analysis, the appropriate word length and integer length are chosen to quantize the model effectively.

Instead of statically setting the scaling factor *S* and the zero point *Z* as shown in Equations (Equation 4) and (Equation 5), the LSQ quantization method [26,27] learns these parameters during training. This approach optimizes the model weights, scaling factors, and zero points simultaneously to reduce quantization error, leading to improved performance compared to static quantization. While most existing studies have focused on applying LSQ to CNNs, to the best of the authors’ knowledge, there are no prior reports of LSQ being applied to RNNs. In this work, the first application of LSQ to a GRU-based classifier for KWS applications is presented. The proposed method will be described in detail in the following subsection.

### 3.2. Methodology

The terminology of Pytorch 2.6.0 [28] is followed for the GRU implementation: (8)$$\begin{array}{cc} r_{t} & =\sigma (W_{ir}x_{t}+b_{ir}+W_{hr}h_{t-1}+b_{hr}), \end{array}$$(9)$$\begin{array}{cc} z_{t} & =\sigma (W_{iz}x_{t}+b_{iz}+W_{hz}h_{t-1}+b_{hz}), \end{array}$$(10)$$\begin{array}{cc} n_{t} & =\tanh(W_{in}x_{t}+b_{in}+r_{t}⊙\left(W_{hn}h_{t-1}+b_{hn}\right)), \end{array}$$(11)$$\begin{array}{cc} h_{t} & =\left(1-z_{t}\right)⊙n_{t}+z_{t}⊙h_{t-1}, \end{array}$$
where $x_{t}$ is the input data at time *t*, $h_{t}$ is the hidden state at time *t*, $h_{t-1}$ is the hidden state at time t−1 or the initial hidden state at time 0, and $r_{t},z_{t},n_{t}$ are the reset, update, and new gate, respectively, σ denotes the sigmoid function, and ⊙ represents the Hadamard (element-wise) product. The learnable weights and biases are divided into input-to-hidden and hidden-to-hidden parts.

Figure 7 shows the data flow of a GRU unit with fake quantization nodes (FQNs) inserted for both activation functions and weights in the quantization-aware training (QAT); bias terms are omitted for the sake of simplicity. FQNs with LUT-based approximation are placed in the reset, update, and new gate outputs. Using a shared LUT across multiple GRU layers for both sigmoid and hyperbolic tangent (*tanh*) activations in future hardware deployment is expected. Since *tanh* function can be derived from the sigmoid value using the relation tanh(x)=2·σ(2x)−1, this approach improves hardware efficiency. The behavior of these nodes is described in Algorithm 1, with an 8-bit LUT covering the range [−128, 127]. During forward propagation in QAT, the de-quantized sigmoid or *tanh* tensors are passed, while in backpropagation, gradients are computed using the Straight-Through Estimator (STE) [29].

FQNs with learnable parameters are inserted for weight tensors, the hidden state ($h_{t}$) output, the feature input tensor in the first GRU layer, and the final fully connected classification layer output, as depicted in Figure 8. Their behavior is described in Algorithm 2.

During training, in the forward pass, these learnable FQNs take the floating point input tensor x and return the de-quantized tensors using the following equations: (12)$$\begin{array}{cc} x_{q} & =\mathrm{clip}(\mathrm{round}\left(\frac{x-Z}{S}\right),Q_{min},Q_{max}), \end{array}$$(13)$$\begin{array}{cc} x_{dq} & =x_{q}\cdot S+Z. \end{array}$$To balance computational efficiency and model performance, the quantization zero point *Z* is omitted for weight tensors (Z=0) but retained for activations. This choice is based on an analysis of the pre-trained full-precision model, which reveals that weight distributions are predominantly symmetric, whereas activations exhibit greater asymmetry.

According to [26], the scaled gradient for the learnable quantization step size *S* is calculated as $s\_grad\_scale=1/\sqrt{Q_{max}\cdot x.numel\left(\right)}$, with x.numel() representing the total number of elements in the tensor x. This formula ensures that, for different step sizes across various tensors and layers, the ratio of update magnitude to parameter magnitude remains consistent. This consistency is crucial for stable training and good convergence. The grad_scale function in Algorithm 2 returns the step size to quantize the tensor *S* during the forward pass, while providing the properly scaled gradient during the backward pass to update *S* itself. In contrast, the quantization zero point *Z* in the work is updated directly using the learning rate. However, this can be further improved by adopting an optimized updating mechanism similar to *S*, as proposed in [27].
**Algorithm 1:** Quantization for Sigmoid and Tanh with LUT Approximation
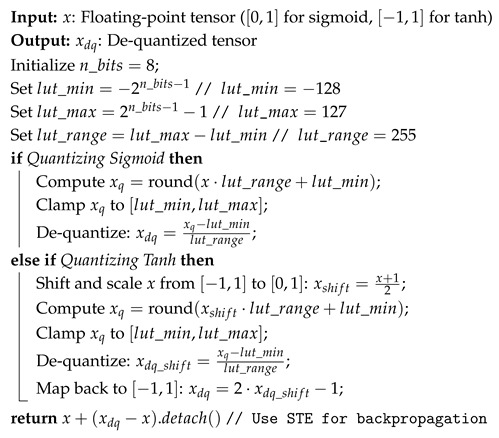


**Algorithm 2:** Learned Step Size Quantization (LSQ) with Gradient Scaling

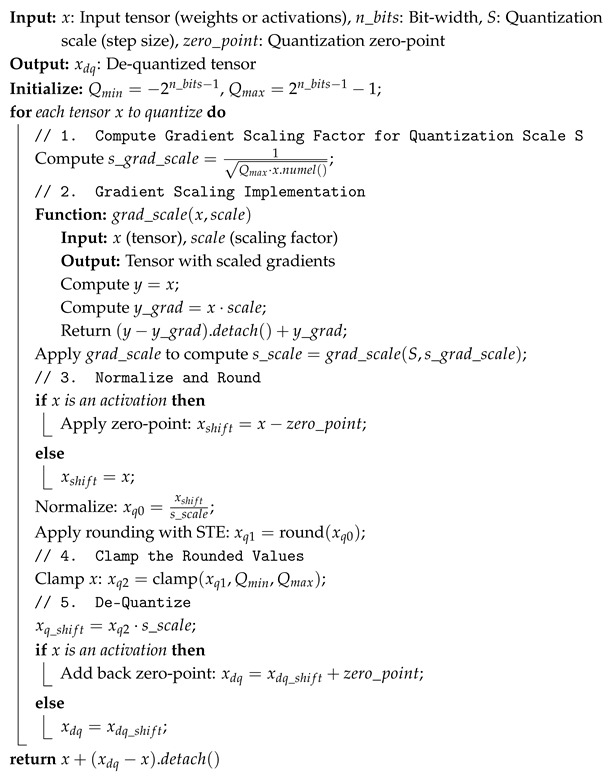



## 4. Training, Performance Results and Comparison

### 4.1. Data Preparation

The Google Speech Command Dataset v2 (GSCSv2) is used to train and evaluate the proposed neural network [30]. The dataset includes 35 words and consists of 105,829 one-second (or less) audio clips recorded by 2618 speakers. The downloaded dataset (http://download.tensorflow.org/data/speech_commands_v0.02.tar.gz, accessed on 23 August 2024) comes with already partitioned training, validation, and test sets (http://download.tensorflow.org/data/speech_commands_test_set_v0.02.tar.gz, accessed on 23 August 2024), such that the keywords spoken by a given speaker always remain in the same set. Besides, the dataset includes several minute-long background noise files. The KWS model is trained for a 12-class classification task. Ten words are selected as the desired keywords and the rest 25 words are grouped into the “Unknown” class. The additional “Silence” class includes multiple 1 s clips randomly sampled from the background noise files. The training set consists of 38,863 samples, including those from the 10 target keyword classes, 4044 “Silence” samples, and 4050 “Unknown” samples. The “Silence” samples are randomly generated from background noise tracks, with volume levels uniformly distributed between −30 dB and 0 dB. The “Unknown” samples are selected from non-target words, ensuring equal representation of each word. The validation set, based on the official partition, contains 4503 samples, with an additional 400 samples each for the “Silence” and “Unknown” classes. The standard test set is used without modification, containing 4890 samples, with approximately 400 samples per class across the 12 classification categories. This results in a training, validation, and test set ratio of roughly 8:1:1, with balanced occurrence of each class within each set.

### 4.2. Training

First, a full-precision model is built using the Pytorch 2.4 framework and trained with Optuna [31] for hyperparameter optimization. The Tree-structured Parzen Estimator (TPE) sampler and median pruner are employed in Optuna, with overall validation accuracy as the optimization metric. The model is trained with the AdamW optimizer [32], cross-entropy loss function, and the ReduceLRonPlateau learning rate scheduler.

The hyperparameters search space includes the following:Number of epochs: 20 to 150;Batch size: 16 to 256;Learning rate: $10^{-5}$ to $10^{-1}$;Weight decay: $10^{-6}$ to $10^{-2}$;GRU dropout: 0.0 to 0.5;Learning rate scheduler decay factor: 0.1 to 0.9.

One hundred Optuna search trials with 30 trials are set for early stopping. The final trained full-precision model achieves an overall accuracy of 92.33%.

To mitigate performance degradation during model quantization, a two-step quantization strategy is adopted, where activation functions and weights are quantized separately. First, the activation functions are quantized while keeping weights in full precision, as activation quantization has a more immediate impact on inference accuracy than weight quantization. This activation-only quantized model is initialized using the parameters of the full-precision model. Next, this activation-quantized model is used to initialize the fully quantized model, where both activations and weights are quantized. This progressive quantization approach enhances training stability, allowing the quantized model to converge faster and reducing the required training epochs.

After initializing the quantized model with pretrained weights and biases, a forward pass over the entire training dataset is performed to estimate the activation range using a moving average window. The activation quantization parameters are initialized using Algorithm 3, following a static determination method, as defined in Equations (Equation 4) and (Equation 5). The weight quantization step size *S* is initialized using the Kaiming He method [33], ensuring that weight values are well distributed within their bit-width constraints, as described in Algorithm 4. For weight tensor quantization, the zero point is set to 0 to maintain symmetry, based on observations of trained full precision model weight distributions.

For both the activation-quantized and fully quantized models, the Optuna search space is minimized by tuning only the learning rate and weight decay, while keeping other hyperparameters constant. This decision is based on some observations from Optuna hyperparameter tuning during full-precision training, where these parameters have had the most significant impact on performance.

Through experiments exploring various weight and activation function bit-widths in QAT (Figure 9), the final model adopts 4-bit weights and 8-bit activation functions (W4A8) for GRU layers, offering a good trade-off between performance and complexity. During the experiments with 6-bit activation functions, input features are kept at 8-bit to avoid excessive information loss. The final classification layer uses 8-bit weights and activation functions, consistent with findings in [26] and suggesting that a higher precision in the output layer benefits overall performance. Notably, the LSQ-based QAT approach enables aggressive quantization with minimal degradation, e.g., W2A8 and W2A6 models still achieve around 90% accuracy, demonstrating its effectiveness.
**Algorithm 3:** Initialization of Activation Quantization Step Size and Zero Point
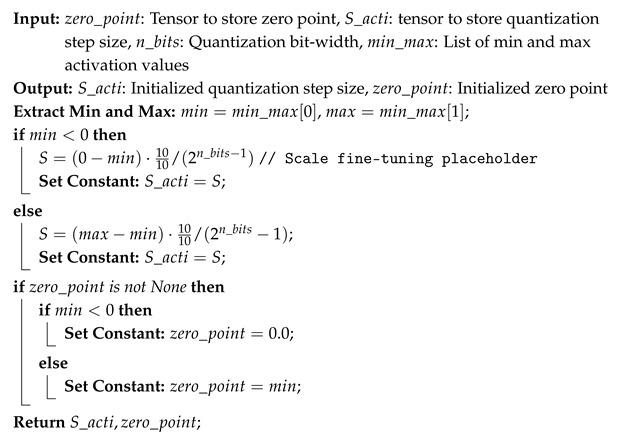


**Algorithm 4:** Initialization of Weight Quantization Step Size

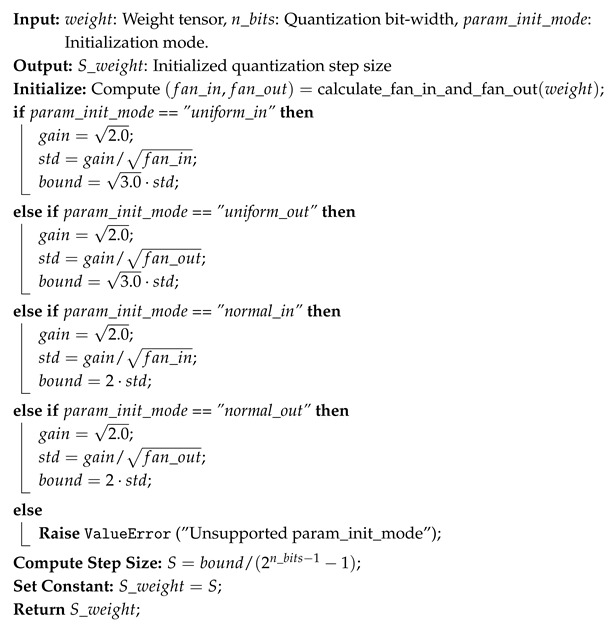



### 4.3. Results and Discussion

#### 4.3.1. Classification Results

The final quantized KWS model achieves an overall accuracy of 91.35% on the 12-class GSCDv2 dataset (10 keywords + “Unknown” and “Silence”), with an accuracy loss of less than 1% compared to the 92.33% achieved by the full-precision model. This result demonstrates that the quantization strategy, based on LSQ and fine-tuning techniques, effectively preserves model performance while significantly reducing computational complexity. Figure 10 shows the true positive rate (TPR) confusion matrix for the 12-class classification task. The quantized model performs well across most classes, with TPRs exceeding 90% for most keywords, highlighting the effectiveness of the proposed KWS system. However, despite achieving 100% TPR for the “Silence” class and performing strongly on words such as “yes” and “right”, which have longer durations and distinct spectral peaks, the model struggles with short-duration words or those with diffuse energy distributions, such as “no” and “up”. The most challenging class is “Unknown”, which includes 25 non-target words, introducing high variability. This variability poses a significant challenge for both the simple analog filter-bank feature extractor (FEx) and the GRU classifier.

The authors would like to advance that the model performance could be even further improved at the expense of increased power and area consumption, as is the case in other works such as [16,34,35]. Potential enhancements include a more advanced feature extraction stage, integrating logarithmic energy scaling or energy derivatives to better capture speech dynamics, and a larger GRU model, capable of learning more complex temporal relationships in the extracted features.

#### 4.3.2. Hardware Resource Estimation

The memory size of the proposed model is estimated by multiplying the number of weights and biases by their respective widths. With a parameter size of 63,372 (excluding quantization parameters), the estimated memory requirement is only 34.8 kB, considering 4-bit weights and 32-bit biases. The number of multiply-and-accumulate (MAC) operations per inference is given by B×(61,440×T+960), where *B* and *T* denote the batch size and number of time steps in the input sequence, respectively. In real-time mode, the model processes one frame per time step, requiring 62,400 MAC operations for inference. In [16] three sets of computational constraints were defined for small, medium, and large microcontroller systems running KWS applications. Based on their analysis, the KWS system of this work easily meets the requirements for small hardware systems, offering a tiny memory footprint and a light computation burden per inference, making it well-suited for low-power edge deployment.

#### 4.3.3. Noise and Analog Impairments Robustness Analyses

To evaluate the robustness of the proposed KWS system, two additional experiments are conducted to simulate AFE non-idealities: (1) computing an input-referred noise and (2) considering the variation of analog filter parameters due to circuit mismatch and PVT variations to finally check the classification performance.

(1)Input-referred AFE noise

Additive zero-mean Gaussian noise is injected into each test audio sample to simulate the input-referred noise of the AFE. The noise standard deviation is dynamically scaled based on the root-mean-square (RMS) value of each input to ensure a fixed signal-to-noise ratio (SNR) per sample. The SNR is swept from 60 dB down to 0 dB to represent varying environmental conditions.

To simulate additive noise at a fixed SNR, the standard deviation of the zero-mean Gaussian noise is computed as follows:(14)$$\sigma =\frac{{\mathrm{RMS}}_{\mathrm{signal}}}{10^{\mathrm{SNR}/20}},$$
where ${\mathrm{RMS}}_{\mathrm{signal}}$ is the RMS value of the clean test audio signal, and σ is the standard deviation of the noise. The additive noise n(t) is generated as follows:(15)$$n\left(t\right)∼N\left(0,\sigma^{2}\right),$$
and added to the original signal x(t) to obtain the noisy input:(16)$$x_{\mathrm{noisy}}\left(t\right)=x\left(t\right)+n\left(t\right).$$

As shown in Figure 11, both the full-precision and W4A8 quantized models demonstrate excellent robustness under moderate noise. The dimensionless input signal is considered. Without noise injection, the accuracy is 92.33% (full-precision) and 91.35% (W4A8). At 60 dB SNR, the accuracy values slightly drop to 92.30% and 91.20%, respectively. Even at 40 dB SNR, the accuracy remains high at 91.92% and 91.08%, respectively. Below 30 dB, the performance gradually degrades, but the W4A8 model performs slightly better than its full-precision counterpart at extreme noise levels (e.g., 10 dB and 0 dB), likely due to a quantization-induced regularization effect.

(2)Filter Parameter Variation (Mismatch and PVT)

To assess robustness against analog impairments, variations in each filter’s center frequency $\omega_{0}$ and quality factor *Q* are simulated using multiplicative log-normal noise. Specifically, each parameter $\theta \in \{\omega_{0},Q\}$ is perturbed according to the following:(17)$$\theta^{\prime }=\theta \cdot \exp\left(\epsilon \right),\;\epsilon ∼N\left(0,\sigma^{2}\right),$$
where ϵ is Gaussian noise applied in the logarithmic domain, and σ is its standard deviation. This formulation ensures that the perturbed values remain strictly positive and that the relative deviations are statistically well controlled. The standard deviation σ is derived from a specified relative standard deviation $r=\mathrm{std}\left(\theta^{\prime }\right)/E\left[\theta^{\prime }\right]$ via the following:(18)$$\sigma =\sqrt{\ln(1+r^{2})},$$
following log-normal distribution properties.

The log-normal distribution is adopted because deep-submicron analog filter designs often rely on subthreshold-biased transistors for ultra-low-power operation, where circuit behavior depends exponentially on underlying normally distributed parameters [36]. Due to the nonlinear nature of the exponential function, log-normal perturbations behave similarly to normal perturbations for small deviations but enlarge them when drifting apart from the mean value.

Two perturbation analyses based on Monte Carlo sampling are performed:Trend exploration: The relative standard deviation *r* of the log-normal perturbation model is modified from 0.5% to 20%. For each *r*, a single random perturbation instance is applied to observe overall classification accuracy trends under increasing levels of mismatch and PVT variations.Expanded statistical analysis: Based on the previously concluded trends, five representative values r={1%,2%,3%,4%,5%} are selected. For each value, 30 Monte Carlo trials are conducted by independently sampling perturbations from the corresponding log-normal distribution. This sampling-based strategy allows the natural emergence of statistical variation patterns, including typical and rare cases, without explicitly constraining the samples to fixed confidence intervals. The resulting accuracy statistics (mean and standard deviation) provide a comprehensive view of system robustness under realistic stochastic parameter shifts.

Figure 12 shows the one-shot perturbed accuracy under varying relative standard deviations *r*. As expected, accuracy gradually decreases as the deviation increases. For variations up to 3%, both the full-precision and W4A8 models exhibit only minor degradation. Under moderate perturbations up to 5%, both models maintain a classification accuracy above 90%, demonstrating excellent robustness. A more noticeable performance decline begins when the relative standard deviation exceeds 8%. Notably, the performance of the W4A8 quantized model remains consistently comparable to that of the full-precision model across all perturbation levels and even slightly surpasses it under extreme conditions. This observation aligns with the results in Figure 11, where quantization presumably implies a regularization effect.

Figure 13 presents the results of 30 Monte Carlo trials for each setting of *r* from 1% to 5%. Both models demonstrate low variance, with standard deviations consistently under 0.25%. Notably, the W4A8 model achieves 90.35% ± 0.23% accuracy at 5% relative standard deviation, reinforcing its suitability for circuit-level deployment with realistic mismatch.

It is worth noting that the perturbation analysis is performed at a behavioral level, without assuming any specific circuit implementation of the analog filters. The deviations in center frequency ($\omega_{0}$) and quality factor (*Q*) are meant to model a combination of possible analog impairments, including process variations, device mismatch, and long-term aging effects. The exact relationship between these parameter deviations and the underlying physical sources depends on the chosen circuit topology, layout strategy, and biasing scheme, which are beyond the scope of this work. The goal is to provide system designers with sensitivity bounds for maintaining classification performance under realistic filter parameter variations.

In summary, the proposed architecture, both in full-precision and W4A8 quantized form, exhibits high robustness to typical AFE circuit imperfections. This resilience supports its deployment in low-power, edge-AI systems. Additionally, the training dataset can optionally be augmented with features extracted from perturbed filter responses (e.g., channel-wise *Q* variation), enabling hardware-aware training for even higher robustness.

#### 4.3.4. Comparison and Discussion

Table 1 provides a comparison between the proposed system and other recent low-power audio inference architectures implemented on silicon. Most of the works focus on KWS using RNN-based classifiers and are evaluated on variants of the Google Speech Commands dataset (v1 or v2). Additionally, a recently published ultra-low-power Spoken Language Understanding (SLU) system is included [37], which targets a different dataset, the Fluent Speech Commands Dataset (FSCD) and classification task (32 SLU classes), but shares key architectural similarities such as 16-channel analog feature extraction and low-bit GRU-based classification. Its inclusion serves to highlight broader trends in AFE design and quantized RNN deployment. The proposed system achieves a competitive accuracy on the 12-class, 10-keyword KWS setup, using a behaviorally modeled analog filter bank and a GRU classifier with 80 units per layer. The proposed LSQ and LUT-aware quantization scheme enables low-bit computation with 4-bit weights and 8-bit activation functions, the smallest bit-width configuration among the listed works, while maintaining comparable classification accuracy (around 91%). This point becomes of special interest towards energy-efficient edge deployment scenarios. Similar accuracy performance can also be obtained using smaller GRU configurations but larger bit-widths, such as in [14,15]. It is also worth noting that simplified arithmetic operations result in a moderately larger memory footprint compared to [12,14,15], due to the higher number of units per layer.

## 5. Conclusions

In this work, a potentially energy-efficient KWS system tailored for edge-computing environments is presented. The proposed architecture integrates a behavior-level analog filter bank for audio feature extraction and a GRU-based classifier optimized through LSQ-based QAT. The filter bank is modeled using second-order band-pass transfer functions to simulate the AFE processing, skipping circuit-level implementation details. The applied LSQ and LUT-aware quantization method ensures low bit-width computation, reducing weights to 4 bits and activation functions to 8 bits in the classifier. The proposed system achieves an overall accuracy of 91.35% on the GSCDv2 dataset for 12 classes with 10 keywords, which means a higher accuracy in comparison to other quantized KWS systems in the literature. The model requires only 34.8 kB of weight memory and 62,400 MAC operations per inference in real-time operation mode. Furthermore, analog impairments analyses demonstrate that the classification performance is maintained under realistic conditions of circuit noise, mismatch effects and PVT variations. The analog nature of the feature extraction, high and robust classification accuracy, tiny memory footprint, and lightweight computational requirements make the proposed KWS system extremely suitable for deployment on low-power edge devices.

## Figures and Tables

**Figure 1 sensors-25-02550-f001:**
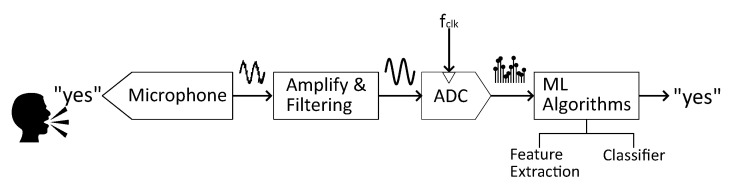
Conventional scheme of a KWS/VAD system.

**Figure 2 sensors-25-02550-f002:**
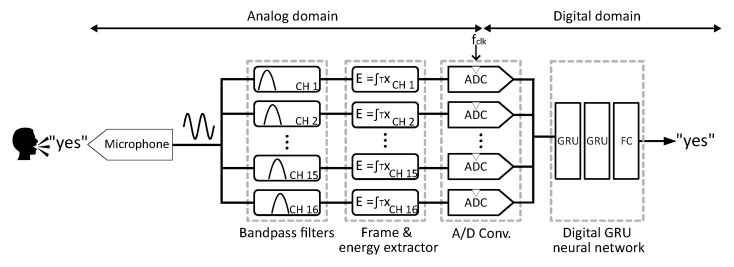
General architecture of the proposed KWS system.

**Figure 3 sensors-25-02550-f003:**
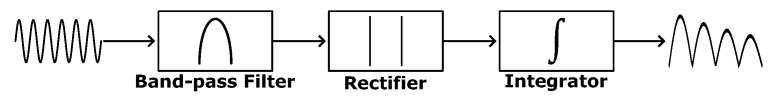
General scheme of the analog feature extraction stage.

**Figure 4 sensors-25-02550-f004:**
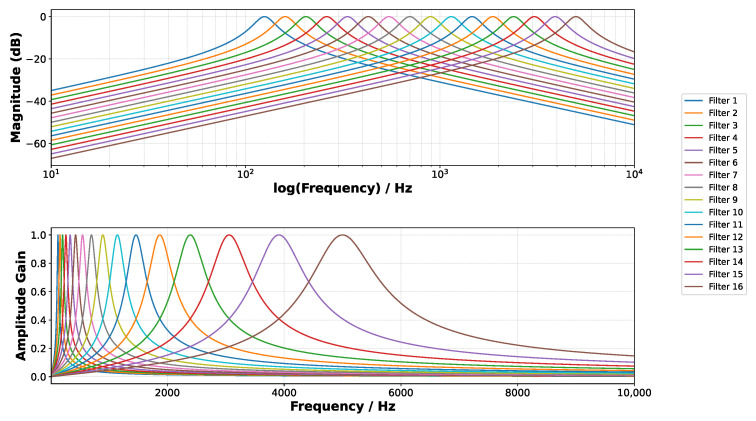
Frequency response of the analog band-pass filter bank in logarithmic (**top**) and linear scales (**bottom**).

**Figure 5 sensors-25-02550-f005:**
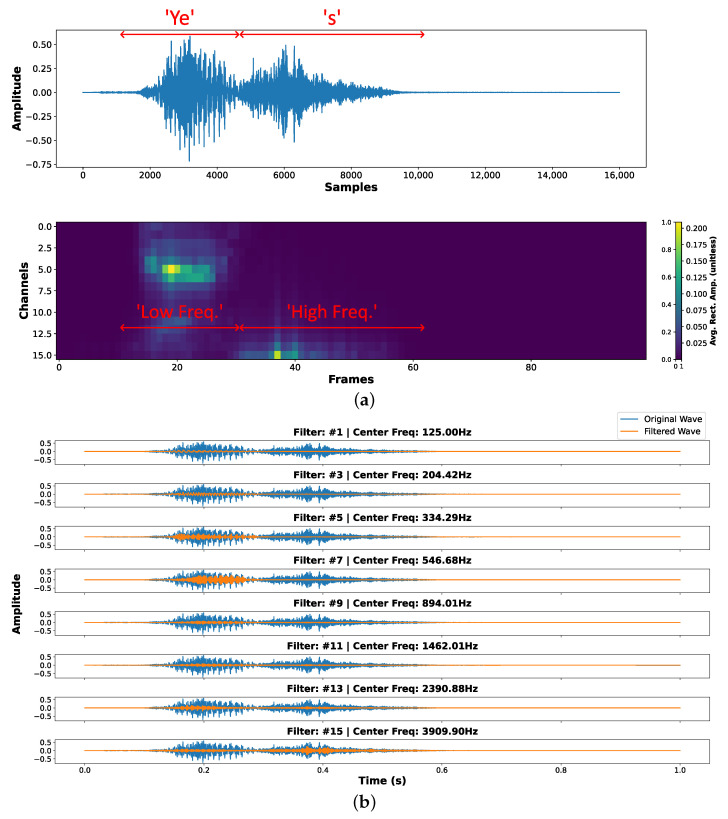
Feature extraction using the keyword “yes” as an example. (**a**) Heatmap of the extracted features, representing the averaged rectified amplitude per frame. (**b**) Comparison between original audio wave and filtered wave for some selected filter channels.

**Figure 6 sensors-25-02550-f006:**
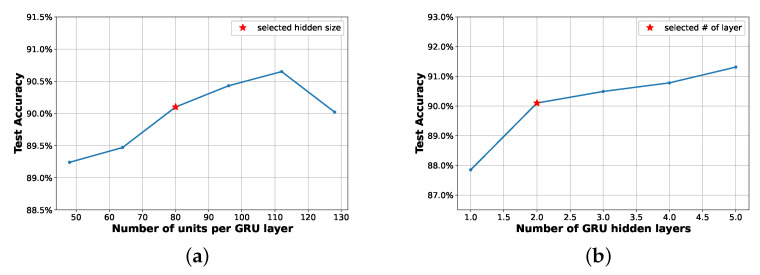
Accuracy on GSCDv2 test set for different number units per GRU layer, and number of GRU hidden layers. Other hyperparameters are kept the same during training. (a) Different number of units per GRU layer, number of layers are set to 2. (b) Different GRU layers, number of units is fixed to 80.

**Figure 7 sensors-25-02550-f007:**
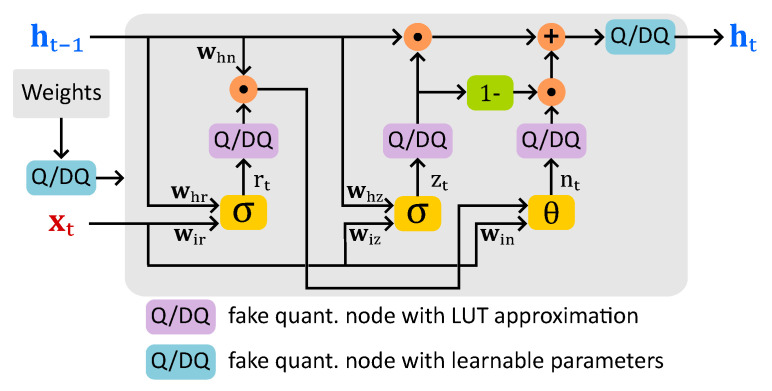
LSQ-based QAT data flow in the GRU unit.

**Figure 8 sensors-25-02550-f008:**
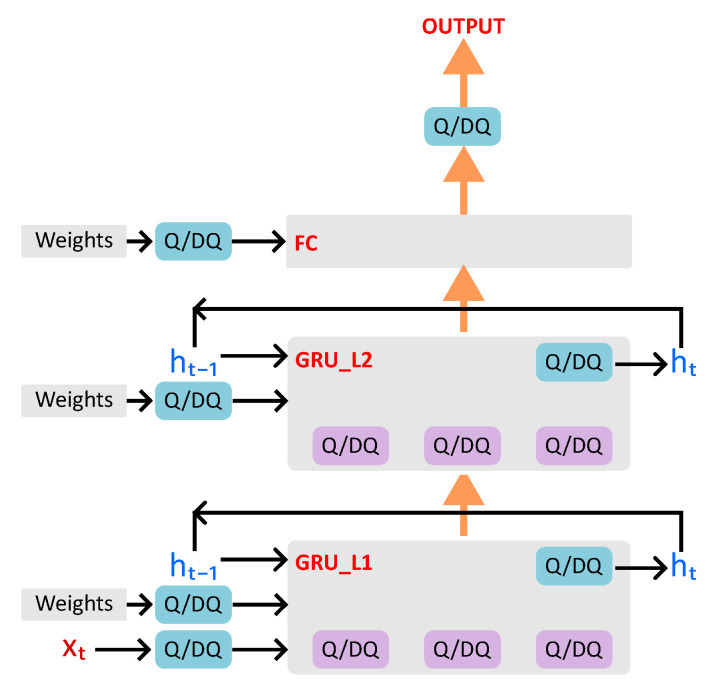
QAT Data flow in the classifier with 2 GRU layers and 1 final fully connected (FC) layer.

**Figure 9 sensors-25-02550-f009:**
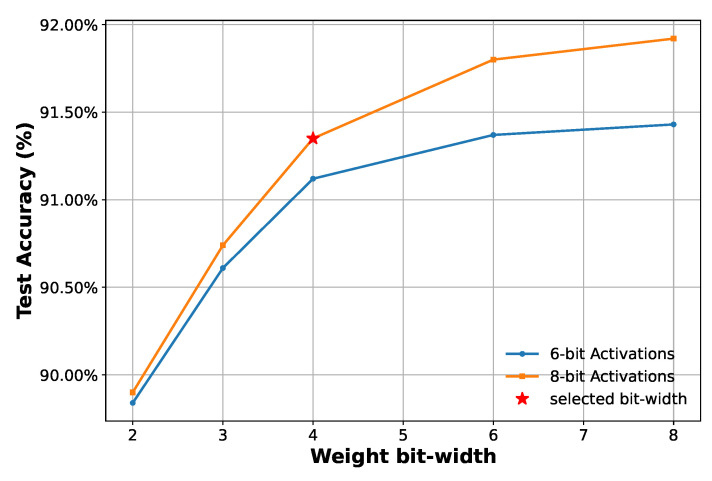
Test Accuracy versus different weight and activation functions bit-widths.

**Figure 10 sensors-25-02550-f010:**
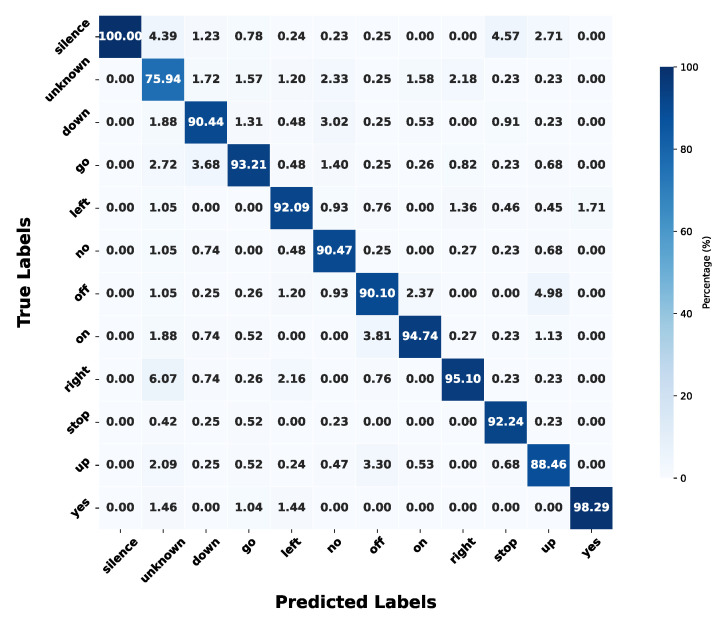
True positive rate (TPR) confusion matrix of the model on the 12-classes.

**Figure 11 sensors-25-02550-f011:**
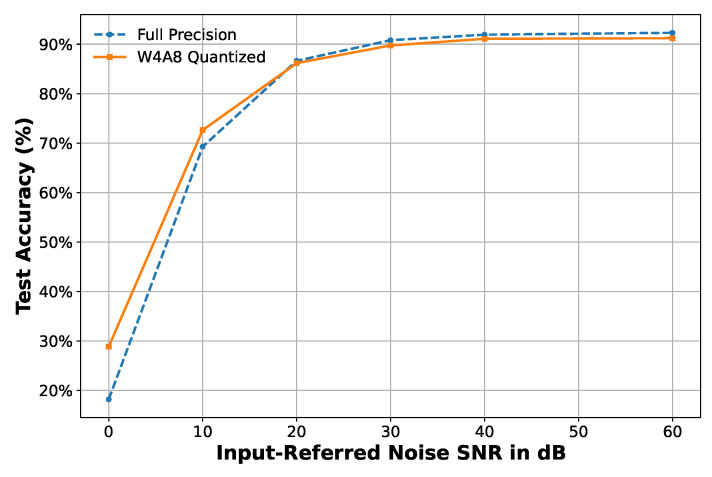
Classification accuracy vs. SNR under injected Gaussian noise. Both full-precision and W4A8 quantized models maintain strong accuracy above 91% at SNR ≥ 40 dB.

**Figure 12 sensors-25-02550-f012:**
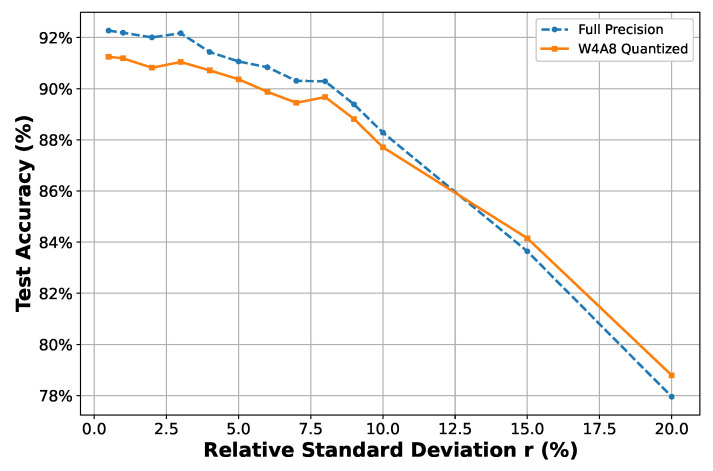
Accuracy under log-normal filter parameter perturbation. X-axis shows the relative standard deviation (r) from 0.5% to 20%.

**Figure 13 sensors-25-02550-f013:**
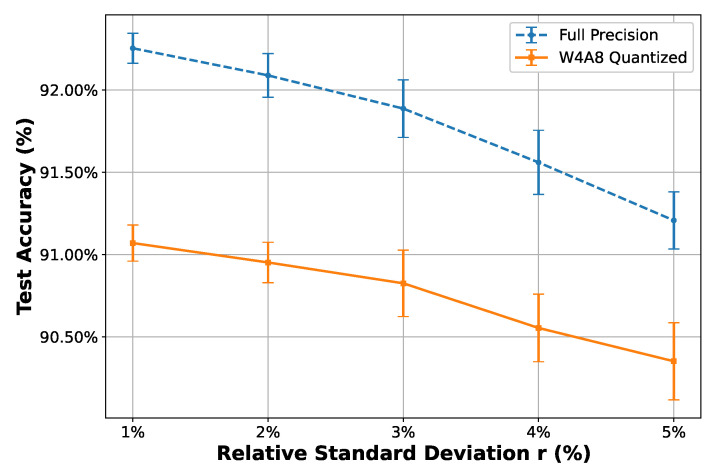
Accuracy under repeated log-normal perturbations of filter parameters (N = 30) from 1% to 5% relative standard deviations. Error bars indicate the standard deviation across trials.

**Table 1 sensors-25-02550-t001:** State-of-the-Art comparison.

	Villamizar TCASI’21 [35]	Kim JSSC’22 [12]	Yang JSSC’23 [15]	Mostafa ISSCC’24 [10]	Chen TCASAI’24 [14]	Zhou ISSCC’25 [37] ^5^	This Work
Feature Ex.	Analog	Analog	Digital	Analog	Digital	Analog	**Analog ^1^**
# Channels	32	16	26	16	10	16	**16**
Classifier	Li-GRU ^2^	GRU	Skip RNN	GRU ^2^	Delta GRU	Delta GRU	**GRU ^3^**
# RNN layers	-	2	1	2	1	2	**2**
Units / layer	-	48	64	128	64	64	**80**
NN Quant.	-	8b w 14b acti.	8b w 12b acti.	-	8b w - acti.	8b w 8b acti.	**4b w** **8b acti.**
NN Memory (kB)	-	24	18	-	24	48	**34.8 ^4^**
Dataset	GSCDv2	GSCDv2	GSCDv1	GSCDv2	GSCDv2	FSCD	**GSCDv2**
# Classes (# KWs)	12 (10)	12 (10)	7 (5)	10 (10)	12 (10)	32 (-)	**12 (10)**
Accuracy (%)	92.10%	86.03%	92.80%	91.00%	89.50%	92.9%	**91.35%**

^1^ Transfer function-based software model, others are silicon-based measurements. ^2^ Off-chip software model w/o quantization. ^3^ Software model w/LSQ and LUT-aware quantization. ^4^ Estimated memory size. ^5^ Targeting an SLU task with 32 classes on FSCD.

## Data Availability

Data are available upon request.

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
