# Peer review of "A KWS System for Edge-Computing Applications with Analog-Based Feature Extraction and Learned Step Size Quantized Classifier"

_sensors, 2025, doi:10.3390/s25082550_

Round 1
Reviewer 1 Report
Comments and Suggestions for Authors
Summary
In the context of low power keyword spotting systems comprising analog feature extraction followed by a neural network, this work presents a method to optimize the quantization of a GRU-type recurrent neural network through a learned quantization step size. Authors show that they achieve a fairly efficient learning in terms of number of units versus classification accuracy with only 16 bands. The main strength of this paper is to describe with good details on the algorithm and their method how they adapted the Learned step size quantization method to the GRU-RNN and made it work to get decent performances. They also have good references on the various flavors of RNN for solving KWS.
General concepts comments
- Authors in [9] show 91% on a real circuit including analog noise as cited in line 56, which is not the case in this work, this applies to other people being probably limited by real electronics effects like mismatch, noise and so on.
- In the table 1, it is not clear that this system is really more efficient than the rest. For example, in ref [13], they are using 10 channels and 1 layer of 64 units of GRU, and they don’t get a much worse results than this work in terms of accuracy on the full GSCD 12 (10).
- Would it be possible to come up with a FoM taking into account the number of weights, number of features, and accuracy? If not, the paragraph underneath that table of comparison could be a bit more measured.
- Would it be possible to add some plots showing the impact of weight and activation quantization on the accuracy, like in Figure 6? Since it is the most important point in this paper, that would add interest to the paper, especially you mention it in line 287.
Specific Comments
- The paper is well written
- Line 90, it is not a gain control then, the filter output is just normalized. Otherwise, it may sound like a dynamic gain control
- The paragraph from line 97 to line 113 is stating the obvious. It could be shortened. It is not an analysis as you look at only 2 phonemes among many phonemes present in the database. It is just an example to illustrate a spectrogram.
- Line 126, I don’t follow the argument about the sensor frequency. Are you talking about the bandwidth? Or the frame rate? If it is the latter, the frame rate is chosen to sample the phonemes rate. It is the nature of the speech.
- Line 135-136 does not sound like a proper reason. Maybe you wanted to compare standard GRU vs GRU with LSQ.
- Line 188, if it is limited, it would be nice to have at least one or two refs with the application they target.
- Line 401 ref [14] – it’s 1.5µW actually
Reviewer 2 Report
Comments and Suggestions for Authors
The paper makes an interesting contribution to mixed-signal pattern recognition systems, now affiliated to ML/AI, that are popular since at least the 90ies of the last century. Predominantly, the idea of low-level preprocessing followed by feature extraction in analog can be found in focal-plane processing and/or vision, nicely summarized in a report of Ali Reza Moini.
However, a major problem with this approach was the significant area consumption, which also the authors mention in their paper. Depending on the technology node employed, this could prohibit the suggested approach as analog computation could take also more power in amplitude domain processing that the application of a highly optimized low-power uC or DSP. At least this has to be carefully analyzed and possibly optimized.
Further, in addition to lack of flexibility with regard to reconfiguring filter or general feature extraction properties, the main issue in analog realization is related to manufacturing tolerances and drift/aging phenomena. The authors in their paper use the word 'analog' more than 35 times, but it seems, currently they regard the analog processing on a behavioral level only, which neglects all these raised issues. Even the filter type and realization, rich of implications for competitiveness of implementation, is not given in detail, e.g., RC, GmC or other, just the order (2nd) and the general band-pass property.
The discussion of the digital part is interesting and the particular learning or adaptive quantization is definitely of value, but the question to which bit resolution quantization should proceed seems not be discussed in the light of the chosen processor or digital solution properties. Or if it has been done it needs a little more detailing and pointing out. In general, it does not become lucid, how exactly the digital part looks like and how the transition of analog stages to digital unit with neural network works (ADC ?).
In the experiments, it does not become clear how often simulation runs have be repeated to mitigate the influence of random effects. Only singular classification values, e.g., 91.35 %, are given in the result presentations in text/tables, mean/sigma would be nice.
Summarizing, the paper has interesting properties, but in the current form it has strong claims on the analog information processing, which either have to be backed up by substantial realization details and assessment, or have to be relaxed w.r.t. to the modeling and investigation level.
Round 2
Reviewer 2 Report
Comments and Suggestions for Authors
The paper has improved a lot, however, it is not quite clear, where the assumption of the possible deviations of filter parameters is rooted, i.e., how are these related to real physical process deviations.
Also, in the paper the effect of 'mismatch' is mentioned inline with the effect of 'PVT', but what is the difference between effect 'P - process' and 'mismatch', in particular' for the particular work ?
